# Implementation of Hospital-Based Supplemental Duchenne Muscular Dystrophy Newborn Screening (sDMDNBS): A Pathway to Broadening Adoption

**DOI:** 10.3390/ijns7040077

**Published:** 2021-11-15

**Authors:** Richard B. Parad, Yvonne Sheldon, Arindam Bhattacharjee

**Affiliations:** 1Department of Pediatric Newborn Medicine, Brigham & Women’s Hospital, Harvard Medical School, Boston, MA 02115, USA; ysheldon@bwh.harvard.edu; 2NCGM, Inc., Raleigh, NC 27606, USA; 3ABDX Consulting, LLC, Raleigh, NC 27609, USA

**Keywords:** duchenne muscular dystrophy (DMD), creatine kinase (CK), targeted next-generation sequencing (tNGS), newborn screening (NBS), newly approved targeted molecular therapies, avoiding delays and early initiation of therapy

## Abstract

Duchenne muscular dystrophy (DMD) is not currently part of mandatory newborn screening, despite the availability of a test since 1975. In the absence of screening, a DMD diagnosis is often not established in patients until 3–6 years of age. During this time, irreversible muscle degeneration takes place, and clinicians agree that the earlier therapy is initiated, the better the long-term outcome. With recent availability of FDA-approved DMD therapies, interest has renewed for adoption by state public health programs, but such implementation is a multiyear process. To speed access to approved therapies, we implemented a unique, hospital-based program offering parents of newborns an optional, supplemental DMD newborn screen (NBS) via a two-tiered approach: utilizing a creatine kinase (CK) enzyme assay coupled with rapid targeted next-generation sequencing (tNGS) for the *DMD* gene (using a Whole-Exome Sequencing (WES) assay). The tNGS/WES assay integrates the ability to detect both point mutations and large deletion/duplication events. This tiered newborn screening approach allows for the opportunity to improve treatment and outcomes, avoid the diagnostic delays, and diminish healthcare disparities. To implement this screening algorithm through hospitals in a way that would ultimately be acceptable to public health laboratories, we chose an FDA-approved CK-MM immunoassay to avoid the risks of false-negative/-positive results. Because newborn CK values can be affected due to non-DMD-related causes such as birth trauma, a confirmatory repeat CK assay on a later dried blood spot (DBS) collection has been proposed. Difficulties associated with non-routine repeat DBS collection, including the tracking and recall of families, and the potential creation of parental anxiety associated with false-positive results, can be avoided with this algorithm. Whereas a DMD diagnosis is essentially ruled out by the absence of detected *DMD* sequence abnormalities, a subsequent CK would still be warranted to confirm resolution of the initial elevation, and thus the absence of non-DMD muscular dystrophy or other pathologies. To date, we have screened over 1500 newborns (uptake rate of ~80%) by a CK-MM assay, and reflexed *DMD* tNGS in 29 of those babies. We expect the experience from this screening effort will serve as a model that will allow further expansion to other hospital systems until a universal public health screening is established.

## 1. Introduction

Duchenne muscular dystrophy (DMD) and Becker muscular dystrophy (BMD) are X-linked recessive neuromuscular disorders caused by variants in the dystrophin (*DMD*) gene. DMD affects predominantly male infants and is the most prevalent pediatric muscular dystrophy with an incidence of 1 in 3500–5000 male births, thus affecting 300,000 children and young adults worldwide, and about 15,000 in the USA [1]. BMD is less common, with an incidence of 1 in 25,000. Of all DMD cases, 65% are due to a large multi-exon deletion clustered in two hotspot regions, and the rest are due to point mutations and duplications [2]. One-third of DMD cases result from de novo mutations in maternal carriers or affected males. The dystrophin gene is large, comprised of 79 exons and 8 tissue-specific promoters distributed across 2.5 Mb of genomic sequence [3]. Dystrophin variants lead to the absence or partial function of the intracellular sarcolemmal dystrophin protein, which normally protects muscle cells from contraction-induced muscle damage [4]. Principal clinical manifestations of dystrophin dysfunction, which include proximal muscle weakness, respiratory insufficiency, and cardiac failure, are progressive and result in death in the second or third decade of life [5]. Dystrophin dysfunction can also cause developmental delays and impaired learning. DMD typically presents by 5 years with clumsiness, weakness or delays in the achievement of motor milestones, clumsiness, or weakness. BMD typically presents between 5 and 15 years of age [6]. The average diagnostic delay between the first-time parents express concerns to a primary care physician to the time of a confirmed diagnosis is 2–5 years. Therefore, the onset of symptoms may precede diagnosis by years. DMD often goes undetected until late pre-school years, when difficulty keeping up with peers is first observed. Gradual weakness and skeletal muscle wasting then ensues, leaving the affected wheelchair-bound by age 11, with most affected individuals only surviving into the late 20s. It is not uncommon for parents to be accused of child abuse due to bruises that result from falls resulting from muscle weakness. Progression and regression create an emotional and psychological burden on the family [7]. A variety of interventions can protect the respiratory system including antibiotics, vaccines, and other ancillary methods [8]. With the prolongation of life with respiratory support, the decline in cardiac function associated with dilated hypertrophic cardiomyopathy is unmasked [6].

Mandatory state newborn screening (NBS) in the US is restricted to disorders for which pre-symptomatic recognition offers potential for a specific treatment that confers a tangible benefit and for which there is political support [9]. Access to rare genetic disease testing is complicated by ordering complexity, high costs, and an obstructive reimbursement process, leaving a gap in its utilization. Universal NBS is one way to resolve this gap in the health care delivery model and reduce disparity. Many Mendelian disorders are not identified through NBS, with the number of diseases far exceeding the number currently screened. The NBS infrastructure and process, now in place for over 50 years, includes (a) collection of a high-quality DBS samples, (b) laboratory analysis on delivery to a centralized lab, and (c) notification and follow-up of positive results. 

DMD therapies are in various stages of development [10]. Recent FDA-approved exon skipping therapies for DMD are appropriate for 30% of the DMD population based on variant-specific mechanisms of action. However, despite being the most common muscular dystrophy, and availability of improved screening and diagnostic techniques, and novel US Food and Drug Administration (FDA)-approved treatment options, DMD is not screened for at birth, and treatment is not initiated until a later symptomatic diagnosis has been made. As clinical trials have generally been conducted in patients aged 6–15 years, optimization of earlier interventions on outcomes remains relatively unexplored. Universal DMD NBS would facilitate evaluation of the efficacy of earlier interventions with these new therapies. As more disease-modifying therapies (including mini/micro-dystrophin gene therapies) enter clinical trials, additional FDA approvals are anticipated. Adeno-associated virus (AAV)-based therapies under development could potentially be administered at younger ages, prior to the development of anti–AAV antibodies, at an age where lower quantities would be needed, owing to weight-based dosing, and perhaps with the introduction of immunomodulation therapies as in Pompe enzyme replacement therapy [11]. It is hoped that gene therapies will one day treat or even cure DMD and allow people with the disease to live longer with an improved quality of life. Unfortunately, the benign adeno-associated viruses (AAVs), which traditionally delivered the healthy version of the intact dystrophin gene into cells, would mostly end up in the liver rather than muscle, as is the case for gene therapy of many other muscle-wasting genetic diseases. MyoAAVs (a family of viruses so-called because they hone in on muscle fibers called myotubes) can deliver therapeutic genes to muscle at much lower doses—up to 250-times lower than what is needed with traditional AAVs. Although this approach has not yet been attempted in humans, animal studies suggest that MyoAAVs largely avoid the liver, raising the prospect for more effective gene therapies without the risk of liver damage and other serious side effects [12].

### Duchenne Screening Test Development and Implementation

Most assays for CK are too cumbersome or impractical for a population-based NBS that uses a DBS-based assay for analytes. The first practical DBS-based CK enzyme activity test was developed in 1975 [13] and used butterfly bioluminescence (luciferin and luciferase). This method was piloted in Germany [14] and France [15]. An electrophoretic technique for measuring skeletal muscle CK-MM isoenzyme levels from DBS was also developed, and screening began in Belgium based on this method [16]. Routine screening programs were then initiated in New Zealand, Wales, France, Cyprus, and Manitoba [17,18,19,20,21,22]. A fluorescence screening test for CK enzyme activity was subsequently developed and replaced the bioluminescence test in a number of screening programs [23]. In congenital muscular dystrophies such as Duchenne, creatine kinase (CK) elevation alone does not confirm a Duchenne diagnosis, as CK can be elevated for other non-specific reasons, such as birth trauma, sepsis, or other muscular dystrophies or conditions [13,24], and confirmatory genetic testing is needed.

In the US, DMD screening was established in 1986 as part of a voluntary Supplemental Newborn Screening Program at West Penn Hospital and Magee-Women’s Hospital [20]. It was offered at no cost and supported in part by a grant from the Muscular Dystrophy Association (MDA). The program used informed refusal. The first-tier CK cut-off was 500 IU/L and if elevated, the sample was rerun. If still elevated, a repeat DBS was requested for CK isoelectric focusing to rule out CK BB isoform as the source of CK elevation and for DNA deletion analysis. Presumptive positives were referred to an MDA clinic for evaluation and possible muscle biopsy if the CK remained elevated and the deletion analysis was negative. All newborns in Pennsylvania and Puerto Rico were screened, as well as in hospitals in Texas, New York, and South Carolina. In total, 623,581 newborns of both sexes were screened, and 59 cases were detected. Recently, a pilot DMD NBS program was carried out in Ohio using CK enzyme activity for a first-tier (37,649 newborn males) and second-tier MLPA-based detection. Six DMD cases (exonic deletions) were identified, as well as other congenital muscular dystrophies caused by variants in *DYSF*, *SGCB*, and *FKRP* [25], thus establishing a second-tier DMD NBS algorithm. More recently, in the Wales program, Moat et al. observed false-negatives due to instability of the CK enzyme that lead to the development of a CK-MM immunoassay [26,27]. The US FDA approved a CK-MM GSP-based immunoassay kit in late 2019 [28]. Recently, this product/assay was piloted in New York State to screen up to 36,000 infants.

Genetic testing is currently used as the next step in the confirmation of a Duchenne diagnosis. Whereas the use of biochemical testing as a first-tier test is the mainstream approach, in recent years, a second-tier test based on tNGS has gained popularity as an option for NBS programs. Since 2009, Whole-Exome (WES) and targeted NGS (tNGS) panel-based DNA sequencing has allowed for significant advances in the understanding of Mendelian Genetics in disorders such as DMD, while contributing to advances in diagnostics [29] and NBS [30]. As precision drugs are developed for genetic disorders such as Duchenne, the utilization of these NGS tools is increasingly valuable to NBS algorithms undertaken by public health laboratories (PHLs). For example, a second-tier analyte/tNGS algorithm is now in use to screen for Pompe disease [11]. Therefore, we chose the best-in-class assays to design a second-tier system of analysis for newborn screening for DMD. This path for NBS minimizes false-positive testing using predetermined DBS CK-MM cut-off levels and tNGS for the identification of *DMD* variants on a single sample.

## 2. Approach and Methods in Hospital-Based Screening Implementation: A Demonstration Project

Between 2018 and 2019, we started developing the concept of a supplemental duchenne muscular dystrophy newborn screen (sDMDNBS). That led to the start of our program at Brigham and Women’s Hospital (BWH), Boston, MA, supported by a gift from the non-profit CureDuchenne. We designed a process based on parental education and consent for a genetic test, the collection of quality DBS samples, off-site laboratory analysis, the return of results to the baby’s hospital Electronic Medical Record EMR), notification of the parents and primary care provider (PCP), and the arrangement of follow-up as appropriate. We concluded that an optional supplement to state-mandated NBS would be the most cost-effective approach to implementation. Our demonstration project would require: (a) parental interest to screen their baby for the disorder, (b) parental education regarding the disorder and the screening test, (c) parental consent for a genetic send-out test, (d) the collection of an additional DBS on a separate collection card, and e) a process to integrate the above into normal clinical care. With FDA approval of a CK-MM immunoassay in late 2019, we posited that this platform would be most readily adopted in the NBS community and care continuum as the first tier of the supplemental DMD NBS algorithm. The second tier NGS techniques (e.g., WES), based on DNA extracted from DBS, had been well developed in our hands and could direct genotype-differentiated treatment for Duchenne (and other conditions if required) on the same initial sample without need for redraw. Whereas DMD is not a time-critical disorder, prolonged turnaround time could engender parental anxiety if the results were not available within a reasonable time- frame. Therefore, we opted to inform parents of the result return time window as part of the education process. Because our group had previously developed NBS-specific requirements for DBS-based tNGS/WES [11,30,31], we focused on tailoring these requirements for DMD NBS. With consideration of the importance of the cost of testing to become implementable for widespread public health-based DMD NBS, we initially set a cost target (inclusive of labor) for the first-tier CK assay at USD 5/sample. Second-tier sequencing costs (at a 1% referral rate and a cost of USD 300 per genomic test) spread over the population would add to USD 3 to the cost of each sample, for a total cost of USD 8 per infant tested.

### 2.1. A Hospital View: The Process for NBS

The state-mandated NBS performed on newborns at BWH is typical of other US hospitals. Although simply described as NBS, obtaining the sample and transporting it to a PHL in a timely fashion is a very complex process starting at the moment of birth. Understanding the details of this process is critical, as our sDMDNBS program piggybacks on existing elements in place as an efficient approach for obtaining consent and collection of a separate sample to pass through the DMD screening algorithm without disrupting the process involving the collection of state-mandated sample send-outs to the New England Newborn Screening Program laboratory (NENSP) facility. Our ability to address the challenges and concerns of all stakeholders prior to implementation guided the design of the infrastructure for our program. All babies born at BWH undergo collection of a heel stick DBS at 24 h after birth by a nurse directed phlebotomist. The samples are sent to NENSP for a state-mandated panel of 35–50 disorders. Each state in the US has unique regulations, processes, and testing facilities, and each birth hospital develops its own unique process to carry out those mandates based on their own infrastructure and resources. The process for ordering, sample collection, sample transportation, testing, return of results, and follow-up required dozens of steps, each of which could create an obstacle to our supplemental program. From prior experiences in attempting to launch DMD NBS programs at other hospitals in different states, we had noted that building such a program at any independent hospital was extremely difficult, with unique barriers at each site stalling the implementation. Further, the general adoption of this approach could differ in various US hospitals depending on its priorities, as well as its governing and regulatory policies. Nevertheless, there may be similarities in the framework which may allow any hospital to integrate such a supplemental program into their newborn care. Mapping this process can be accomplished by following the baby and sample from birth to sample shipment. Key elements include admission order templates; introductory materials for parents to explain the disease and screening algorithm (including how to integrate/disseminate this information with other parent materials); the relationship of an offer for supplemental screening to the medical and nursing staff interactions that introduce newborn screening to parents; mechanisms for documenting consent for performing a clinical genetic test (required by any CLIA certified laboratory that might perform NGS); the steps involved in ordering the supplemental screen through the EMR in a way that allows tracking and signoff, integration into the laboratory result portion of the EMR; sample collection details (nursing or phlebotomy, transportation of the sample from the patient to the hospital laboratory, login and pre-send out of samples in the hospital laboratory, development of sample manifests for bulk shipment to the outside laboratories that perform the CK and NGS testing, mechanisms for shipping under rapid and stable conditions, details of result return into the EMR, notification of parents and providers of normal and abnormal results, and construction of a follow-up program by appropriate subspecialists for diagnostic confirmation, treatment planning and monitoring). In Massachusetts, we have the benefit of a state NBS program that, for many years, has offered an optional pilot panel of disorders that are being pre-evaluated for integration into mandated screening. Our medical and nursing staff already have standardized procedures in place for offering optional state NBS pilot testing (which requires consent) to parents. Because this interaction was already in place, we were able to take advantage of attaching the additional offer of a supplemental test. By putting this burden on the primary in-hospital caregivers, the need for adding personnel specifically for offering the sDMDNBS was avoided. In our experience, supplying written materials alone, such as brochures, is not adequate to ensure a strong parental uptake response, as they are typically overwhelmed with a variety of written materials that are difficult to review during the post-partum in-hospital period. The introduction of the program verbally, on a scripted protocol by the care staff, ensures the parents are aware of the program and can ask questions prior to obtaining a specific yes/no response from each family.

### 2.2. CONCEPT: Piggybacking onto the Infrastructure of State-Mandated NBS

The supplemental duchenne muscular dystrophy newborn screening (sDMDNBS) program is currently active at BWH. It was preliminarily developed by meeting with all stakeholders identified in the above processes and confirming their support in operationalizing the program. Over a 6-month period, meetings were held with the Newborn Intensive Care Unit (NICU) and Well Baby Nursery (WBN) medical and nursing leadership (and then, with buy-in, their staff), laboratory medicine, form approval committees, printers, filter paper suppliers, administrators that will pay vendors, phlebotomy, laboratory control, information systems, outside laboratory staff, and notification of the state newborn screening program. We did not initiate these stakeholder meetings until the funding support needed to manage the program and pay for the testing was identified. In our case, this was through a gift from CureDuchenne. The final process for supplemental hospital-based (to be distinguished from state public health-based) NBS programs was subsequently developed after all stakeholders had been involved (Figure 1). Implementation of the developed process was carried out by a part-time effort of a physician champion (RP), an experienced nurse program manager (YS), and a laboratory champion (AB), who were collectively responsible for the generated appropriate forms, SOPs, protocols and workflows, to allow for consistency at our own institution and the potential for translation to other birth hospitals. We opted to offer sDMDNBS to the parents of all newborns regardless of sex given that DMD can, although less commonly, affect females. Before our formal launch, we pre-launched by releasing 100 parent information packages to post-partum parents. In total, 83 samples were obtained. These were batched (stored at −20 °C) and sent to the testing lab. A manifest for the batch was reviewed with the lab, ensuring all orders had appropriate consent and satisfactory samples. CK results were returned in ~1 week and uploaded to the EMR. Once a complete cycle from collection to return of results had tested all steps of the process, we were able to make adjustments prior to a full launch. The first samples above the CK cutoffs were returned from WES 3 weeks later. Laboratory protocols were set for how the results for upload were described for visualization by parents and clinicians, ranging from normal (CK in-range) to abnormal (elevated CK and pathogenic variant detected). For the first 3 months (July–September of 2021), the program yielded results on ~1500 newborns with 80% uptake by parents who were offered a sDMDNBS at no charge. A weekly meeting was held between the physician champion, program manager, and laboratory champion to review process issues and clarify questions about the samples from specific newborns.

### 2.3. DBS Card Shipment and Stability

As storage and shipping conditions also impact the CK-MM values (although likely less than CK-enzyme) [26,27,28], we, shipped samples overnight at ambient temperature at the beginning of the week in thermally insulated boxes. The DBS filter paper cards were kept in foil envelopes to avoid exposure to excess moisture (humidity) and temperature with a desiccant pouch, as humid conditions can lead to < 80% recovery in 2 days at 21 °C and above. After processing samples, the sample cards (residual spot) were stored at -80 °C.

### 2.4. Two-Step NBS Algorithm for DMD (Pre-Pilot and Pilot Trial)

As we considered the DBS collection process, it was thought parents would feel most comfortable with a single heel-stick to provide samples for both the state-mandated NBS and the sDMDNBS (i.e., it would be more traumatic to the newborn and family to do multiple heel-sticks). We accepted that the blood volume collected at the end of the collection might not be sufficient to fill multiple circles, limiting our collection to one single filled spot (Figure 2). This process was presented to the parents prior to consent so that they understood the intervention and extent of additional blood to be removed from the baby beyond that collected for the state-mandated NBS. Thus, the algorithm planned for the CK assay and tNGS/WES to be performed on a single DBS (roughly equal to 50 ul of whole blood), which typically produces between six and eight 3.2 mm punches (Figure 2). We mock tested this process in pre-pilot studies, evaluating the first-tier testing (CK-MM) followed by tNGS/WES. We switched to the FDA-approved Perkin Elmer (PE) CK-MM assay as it was more likely to be adopted by the public health newborn screening laboratories than our originally proposed home-brew CK-enzymatic assay, which is difficult to implement for DBS and requires extensive validation. We ultimately expect that the PE CK-MM assay is likely to cost closer to USD 5/baby when deployed for screening nearly 4 million babies per year in the USA, similar to the cost of other similar assays (IRT, TSH, 17-OHP) that the company currently markets to PHLs. The final choice of a two-tier algorithm with a fixed cut-off was based on the ability to perform both the first-tier CK-MM and second-tier tNGS on the initial sample to: (a) avoid the need and resources to recall families for a repeat sample for second-tier CK testing prior to NGS, and (b) lower the initial false-positive rate and minimize parental anxiety.

For tNGS/WES, a hybrid capture is performed using a custom Twist Exome capture kit that targets coding regions, splice sites, and clinically relevant non-exonic hotspot regions with the initial clinical analysis being performed on the *DMD* gene. Paired end sequencing is performed using the Illumina NextSeq2000 system. Reads are assembled and were aligned to reference sequences based on NCBI RefSeq transcripts and human genome build GRCh37/hg19. The average depth of coverage across the target is 100X. Approximately >99.5% of the target region is routinely covered at a minimum depth of 20X. FastQC/Fastp is used to assess the quality of the raw data. Variant calls are generated using Samtools for the Burrows-Wheeler Aligner (BWA) followed by GATK based analysis. This test detects >99% of substitution variants and >80% of small insertions and deletions up to 25 bp in length across the genes based on sequencing of NA12878 compared to reference calls from the National Institute of Standards and Technology Genome In a Bottle Consortium. Variants are interpreted in a manner consistent with the 2015 American College of Medical Genetics and Genomics and Association for Molecular Pathology recommendations, and variants classified as being of uncertain significance, likely pathogenic, pathogenic, and other variants, and reported if related to the patient phenotype. Benign and likely benign variants are not included in this report. Given that a majority of DMD cases are due to deletion events (60–70%), we made CNV calls on the WES dataset. In addition, we performed a manual check for coverage for each exon and a flag for no read or low reads. As this is a disorder associated with a gene on the X-chromosome, and males have only one X-chromosome, missing CNV calls in males is unlikely. Females have a very low incidence of the disease but there is a risk that the heterozygous call may be missed. We confirmed multi-exonic calls in samples from affected individuals with DMD and other disorders by other molecular assays. The DNA quality is more likely to impact duplication detection, but this is a general limitation of short-read NGS. This issue may be slightly more pronounced in DBS as compared to whole-blood samples. The primary reason for our confidence in our CNV calling is our dynamic baseline approach in which baselines are created from a pool of thousands of samples to match as closely as possible to the sample under analysis. This significantly improves sensitivity which is often lost in an attempt to ’normalize’ the baseline.

### 2.5. Post-Analytic Workflow sDMDNBS: Result Transmission

The results are returned to the program manager for preliminary sorting (Figure 3). CK results are returned within 1 week. If the CK is below the cut-off, the result is uploaded to the laboratory result section of the baby’s EMR and can be seen by the parent as “normal.” If the CK is elevated, a WES is sent but the results are not available for an additional 3 weeks, so the EMR remains in a pending mode. When the NGS-WES result returns, if there are no *DMD* variants identified, the program manager arranges for a repeat CK via a phone call with the parents, and then releases the NGS-WES report to the EMR. The result is seen by the parents in a way that suggests the CK was elevated but the *DMD* sequencing did not reveal a variant. They understand from the phone call that a repeat CK is being requested to confirm that the value has fallen to the normal range, and that if it is still elevated, a referral will be made to our DMD NBS follow-up clinic for further evaluation. If the CK is elevated and the NGS-WES returns with a *DMD* variant, the physician champion contacts the baby’s PCP and provides the necessary information to be communicated to the family to bring the baby into the follow-up clinic for further evaluation. The follow-up clinic, which exists as a component of the Boston Children’s Hospital DMD clinic, has a protocol for intake of DMD NBS positive babies which involves the confirmation of diagnosis, genetic counseling, discussion of appropriate treatments, and a schedule of close follow-up. Despite our concern about the possibility of our contact regarding the need for CK repeat causing parental anxiety, in the first 15 cases that we have contacted (made by an experienced NICU nurse), there was no immediate suggestion of a negative impact of the information. With one exception, mild anxiety was noted in one mother who was listed as a primary English speaker, but because English was not her first language, and she had a mild problem with comprehension. We strongly suggest the use of a translator if there is any question that this may be an issue.

## 3. Results

### 3.1. Two-Step Algorithm Results (Pilot)

In a 3-month period (June 2021–September 2021), we processed approximately 83–118 newborns per week for a total of 1379 newborns. In total, 80% of approached parents opted to have their child undergo the sDMDNBS. The initial pre-pilot set, we ran a study for approximately 1 week (*n* = 83). The date of collection was between 24-48 h of birth for all collections. During our pre-trial, we realized that there was no available evidence for gestational or birthweight-based PE CK-MM normal cut-off values for premature or NICU infants. As the CK level is proportional to muscle mass, there was concern that gestational age might influence norms in prematurely born infants [32]. We therefore focused our screening on term newborns who were mainly admitted to the WBN. We allowed a choice to screen term babies who were ultimately admitted to the NICU. On receipt of the samples by the testing laboratory, the filter paper cards were assessed regarding how completely the spot was filled and whether there were any other quality-control issues (Figure 4). In one instance, a shipment was delayed from the usual overnight to a 4-day transit during a hot summer month, and we did not see any decay in mean CK-MM levels. The preliminary CK-MM fixed cut-offs were based on the FDA-approved assay normal cutoffs for males at 1080 ng/mL and females at 958 ng/mL, and were performed at the PE facility in Pittsburgh, PA. The initial 83 clinical samples from infants born at 35 weeks gestational age or above in the well-baby nursery had a CK-MM mean = 358 ng/mL, and the SD = 186.8, mean + 2SD (95%) = 732 ng/mL, and mean + 3SD (99.7%) = 918 ng/mL. On pre-pilot we had no referrals, as the highest CK value was 858 ng/mL (Figure 4). Among the 1375 samples screened to date that had a valid CK-MM value, the mean was 385.5 ng/mL, and the SD = 266.7, mean + 2SD (95%) = 800 ng/mL, and mean + 3SD (99.7%) = 1185ng/mL. On average, we had 0–6 samples per week with an elevated CK above the fixed cut-offs. To date, we have processed 29 tNGS/WES (2.1%) over the 1379 samples screened for CK. Our program found only three newborns with CK >2000 ng/mL (2040, 3220, and 3755 ng/mL). One patient with elevated CK (3220 ng/mL) was suspected of another syndrome when evaluated, and a WES study was ordered (see next section). Thus far, the elevated CK samples processed for WES have revealed no variants in the *DMD* gene. The quality control data showed that, over time, most quality-control aspects improved (Figure 5). 

When we additionally consider CK-MM repeat testing after no variants have been detected on tNGS, the residual risk of missing DMD is substantially reduced. If there is no third-tier CK performed after the *DMD* tNGS, there may be a risk for missing disorders other than Duchenne, as may be indicated if the CK remains persistently elevated (see Section 3.2).

### 3.2. A Non-DMD Abnormality Identified in the Exome of a Newborn with Elevated CK at Birth

The parents of a term female requested the sDMDNBS through our standard process. Because of the presence of contractures on the newborn exam (not suspected on prenatal imaging), a CK was drawn as part of the medical evaluation, and CK levels were extremely elevated. Because this was known in advance, and access to a rapid clinical Exome was not assured, a request was made to fast track the tNGS/WES of the DMD screen. The CK elevation was also confirmed on the DBS. A phenotype was provided, and a broader sequence analysis was performed. A WES performed from both the sDMDNBS and a separate clinical Exome were consistent. The analysis was based on Human Phenotype Ontology (HPO) terms derived from the presenting clinical summary that rapidly identified a Chr 16 microdeletion syndrome (Chr 16p11.2).

## 4. Discussion

Establishing universal (all 50 US states) NBS for a new disorder when Wilson and Jungner criteria [33] are met is a lengthy process which may take 5–10 years to accomplish, and for which evidence basis is difficult to collect. This is true for DMD, which has been known to be screenable for 45 years but has not had available FDA-approved therapies, until the past 5–10 years [10]. However, until RUSP approval, there will be a gap in Duchenne diagnoses and adverse consequences. Indeed, over the last 20 years, the mean age of diagnosis (5 years) has not improved, suggesting that at least 600 newborns per year run the risk of not being diagnosed early and thus missing out on early initiation of therapy or choice of participating in clinical trials. We have demonstrated proof-of-concept in the implementation of a hospital-based supplemental DMD NBS program. We believe that we are now ready to expand from screening the newborns at a single birth hospital to offering to assist implementation across our health care network. This would have the potential to increase the offering from 7000 to 17,500 of the 70,000 babies born annually in Massachusetts within weeks of birth.

Algorithms that rely on second or third sample collection before tNGS can be performed are susceptible to “lost to follow-up” (due to changes of address, phone, PCP, failure to answer phone or mail contact, or failure to appear at the PCP office). Collection at a second timepoint is labor-intensive and dependent on the supplemental program staff and private pediatrician rather than the hospital infrastructure with a captive patient. Time may be lost in initiating therapy, waiting for second sample collection, and laboratory testing (most important in time-critical disorders with very early onset). Other benefits of our proposed algorithm include: (a) Orthogonal measurement: There is less bias due to the concurrent use of phenotype and genotype in making positive calls. (b) Clinical benefit: Uses genotype information that is needed to target the most appropriate treatment (e.g., specific exon skipping therapy) and specificity. (c) Simplified Comprehensive Testing: covers exons and point mutations, as well as CNVs. Other predictions, such as X-Linked DCM, or other non-DMD causes can provide guidance to clinicians. (d) Reflex to other muscular dystrophy-causing genes in the setting of persistently elevated CK but no *DMD* variant detected. If the DMD NBS had not been performed, families may be hurt by missing the opportunity to avoid a diagnostic odyssey; and (e) Avoids disparity: as reimbursement policies make it difficult to obtain genomic testing, including programs like Medicaid, and those families without insurance may not benefit at all.

The alleged negative effect on the perception of parents about baby and bonding due to ‘false-positive’ results is minimized in our algorithm [10], as DMD is a highly penetrant condition that can be confirmed by a second-tier tNGS test from the original sample. If a genetic counselor can quickly explain the results to the parents, they will be less likely to have a bonding problem in the setting of false-positive results. Bonding to DMD children can be an issue, but historically, parental attitudes, as well as those of their health care providers, support the value of screening. Our uptake is consistent with what has been historically observed with prior DMD screening efforts in the US. A primary focus of a prior study was whether parents of affected sons perceived the benefits of genetic counseling to be greater than the emotional trauma of discovering their child was affected with a devastating disease prior to symptoms [34]. In total, 80% of mothers of affected boys not screened supported screening (4.0% against; 16.0% unsure), and 69% of parents of affected boys diagnosed through screening supported the program (17.2% against; 13.8% unsure) [34]. Many felt that genetic counseling prevented the potential negative effects of delays between the onset of symptoms and diagnosis, that they had the right to know as soon as possible, that it could prevent birth of affected siblings, and that there were practical and emotional advantages. Previously, there were objections, as definitive treatment was not available, and consequent negative psychological effects, such as rejection, interference with bonding, or overprotection were perceived. The availability of screening and therapy permits informed decisions, including future pregnancies, which could impact the number of affected cases. If screening is carried out to detect affected males and, subsequently, female carriers, it may prevent 25–30% of future cases [13]. Most families of affected boys favored screening on the grounds of reproductive choice and time to prepare emotionally and practically. So far, there is no evidence of a long-term disruption of the mother–baby relationship, anxiety levels in our screened groups that did not subside, or adverse consequences from anxiety or wellbeing scores indicating that a group suffered any disadvantage. We believe that the opinion of parents should be considered in addition to that of experts in the decision-making process of what conditions to include in future screening programs. The adoption of testing in our program demonstrates strong support for newborn DMD screening among unselected parents even today, as well as support among parents of children over the past 25 years.

Unnecessary health care utilization is considered an issue by some [10]. Until effective therapy becomes widely available, the only benefit to the patient, family, and society is more subjective and more controversial. Since 13–15% of newly diagnosed cases are found to have an affected younger brother, genetic counseling, family testing, carrier testing, and prenatal or preimplantation genetic diagnosis allows for more informed reproductive decisions and may potentially prevent secondary cases [18]. If a hospital wants to screen for other forms of muscular dystrophy, that is also possible if the CK-MM testing cut-off is lowered to detect those forms, and WES can identify or rule out any of those conditions.

Timonen et al. [28] reported a mean for CK-MM at 328 ng/mL, with a 95^th^ percentile cut-off at 867 ng/mL and 99^th^ percentile at 1190 ng/mL in the US cohort, which is similar to our observation. The study identified a preterm (gestational age 27 weeks) baby with low birth weight (1200 g) that resulted in a false-negative result (55.2 ng/ mL), and another with a gestational age of 35 weeks and birth weight of 2700 g had a CK-MM concentration of 1100 ng/ mL, i.e., above the used cut-off value [28]. Previous prenatal studies have shown that some affected fetuses do not have elevations in CK [32]. Since the CK-MM is primarily a marker of skeletal muscle damage, premature infants under gestational age of 26 weeks and certain birth weights may need further evaluation to ensure premature Duchenne infants are not missed. Whereas muscle damage may start in utero, less muscle damage and less muscle mass (and thus less indirect markers of Duchenne) may result in false-negatives. Duchenne NBS programs might also benefit from using a lower-cost first-tier molecular screening algorithm for low birth weight and preterm newborns, as well as screening for BMD and female carriers, however this would need further development. Eventually, we expect that commercial-grade FDA-approved assays will be developed by multiple NBS technology companies. In turn, this will drive competition and lower first- and second-tier screening costs, and allow any clinical laboratory of any size (private or public health affiliated) to implement the desired scale.

In the near term, the target NGS target test cost of USD ~100–200 may be achieved using smaller NGS panels if the WES cost is too high. The use of a neuromuscular panel of approximately 100 genes that covers DMD and other muscular dystrophies, including Limb Girdle Muscular Dystrophies, is used in the absence of wide availability of exome sequencing from DBS through commercial vendors. However, other challenges encountered may be necessary yearly upgrades to NGS panels. As the LGMDs are autosomal recessive or dominant disorders, screening for females can enable the identification of variants in autosomal genes or those heterozygous for *DMD* mutation who are carriers and/or manifest a form of the disease, allowing this two-tier method of screening to account for subjects with elevated CK levels.

To date, none of our elevated CK-MM values have resulted in the identification of a *DMD* variant. For neonates with an abnormal neuromuscular exam, we have made a phenotype-directed diagnostic analysis beyond *DMD* on an exome generated due to CK-MM elevation accessible to our clinicians, after obtaining expanded consent for additional WES analysis. One such newborn with a CK-MM value above cut-off and reflexed for tNGS/WES testing had a heterozygous pathogenic Chr 16 microdeletion identified. This finding would not have been detectable using pre-fixed neuromuscular tNGS panels. One gene out of the 14–25 genes deleted in Chr 16 microdeletion is *ALDOA* [35]. Although only one copy is lost, *ALDOA* may have haploinsufficiency effects consistent with the newborn’s phenotype. Aldolase A (ALDOA) is the predominant isoform of aldolase in skeletal muscle and erythrocytes that catalyzes the reversible conversion of fructose-1,6-bisphosphate to glyceraldehyde 3-phosphate. Autosomal recessive mutations in *ALDOA* are extremely rare and cause hemolytic anemia and/or recurrent episodes of rhabdomyolysis, usually precipitated by fever. A homozygous or compound heterozygous mutation or null variants can cause severe muscle pathology and death, as well as overlaps with Glycogen Storage Disorders XII (GSDXII). The episodic or recurrent nature of Alsolase A deficiency points out that triggers due to fever or muscle tension may induce elevated CK. The impact of this enzyme deficiency on the brain may also potentially contribute to neuropsychological aspects of the condition, a feature shared with GSDXII. Thus, elevated CK and the WES finding were able to benefit the newborn with a prompt diagnosis for a highly heterogeneous disease. Given that the WES sequence had already been generated on this newborn and was available for analysis, a rapid diagnosis was possible, and costs could be saved.

There is a risk that screening for CK could lead to the identification of disorders that are not treatable. So far, NBS programs that have utilized screening algortihms with a first-tier elevated CK, and a second CK or gene panel limited to known disorders or genotypes has reduced detection of newborns with initial CK elevations due to muscle trauma related to birth. By increasing the CK enzyme activity cut-off value to 2000 U/L, algorithms have limited the identification of diseases such as LGMDs that do not have approved therapy, although typically these were identified at a much lower frequency [25]. Gene therapies in LGMDs are now entering the early stages of clinical trials. Therefore, both non-specific muscle-preserving treatments (such as steroid treatment), symptomatic treatment, and disease progression-halting therapies may become available. Corrective surgeries or monitoring of cardiac and pulmonary function are also important management approaches. Nevertheless, in our model, these issues are addressed by the referral to a follow-up center of any newborn whose repeat CK remains elevated in the absence of an identified *DMD* variant. 

## 5. Conclusions

In summary, the DBS-based two-step algorithm (CK-MM as the first tier and tNGS/WES as the second tier) is suitable for NBS of Duchenne. We need additional data specific to the gestational age of the baby for CK values, as prematurity and low muscle mass may cause false-negatives. WES can ensure that NBS multi-tier analysis fully reveals the cause of an elevated CK (including non-DMD etiologies). Combining a biochemical first-tier with a molecular second-tier test in the NBS allows us to better understand the disease natural history (genotype and phenotype) and subsequently decide on treatment strategies.

The primary goal of NBS is to identify patients who can be treated to establish significant health gain. Secondary goals, such as shortening the diagnostic odyssey, identifying carriers, and providing information for reproductive options, are not considered appropriate by Wilson and Jungner criteria [34]. However, as *DMD* gene defects or elevated CK levels reflect a wide range of conditions, simply providing biochemical screen positive information alone delays treatment initiation in those phenotypes. Outside of DMD, early genotype information alongside the CK profile may allow prompt identification and treatment initiation as in infantile onset Pompe disease and LGMDs [11]. Several state PHLs have already introduced second-tier DNA sequencing in their NBS algorithms [11]. The second-tier tNGS test for Duchenne can provide rapid and precise information on highly penetrant recurrent pathogenic and de novo variants, avoid false-positives, provide choice for exon skipping therapy, and identify variants associated with DMD and other disease phenotypes. A combination of early detection, close monitoring, and early use of therapeutics such as Emflazacort or other steroids in addition to first-generation exon skipping therapy (PMOs), will likely improve outcomes, but additional data are needed. Consistent with the original intent of NBS, Duchenne treatments should be considered in the newborn phase, but use of the best screening assays and alternative implementation methods must be evaluated to make meaningful impact over the course of life. We demonstrate that hospital-based DMD screening, for now, and until it is adopted by mandated newborn screening programs, is both feasible and sufficient to fill the gap.

## Figures and Tables

**Figure 1 IJNS-07-00077-f001:**
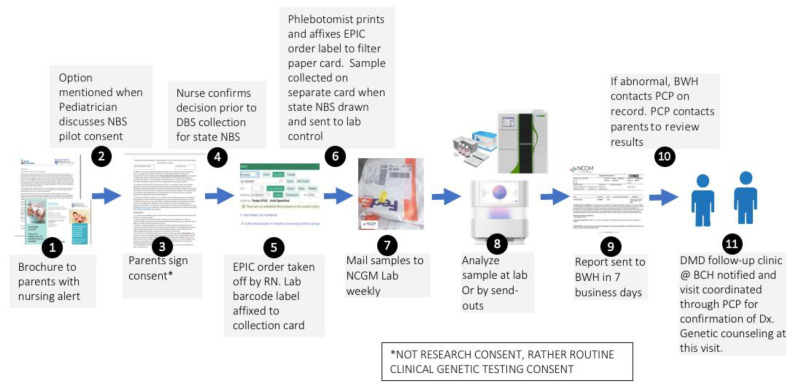
BWH supplemental Duchenne (DMD) newborn screening workflow steps.

**Figure 2 IJNS-07-00077-f002:**
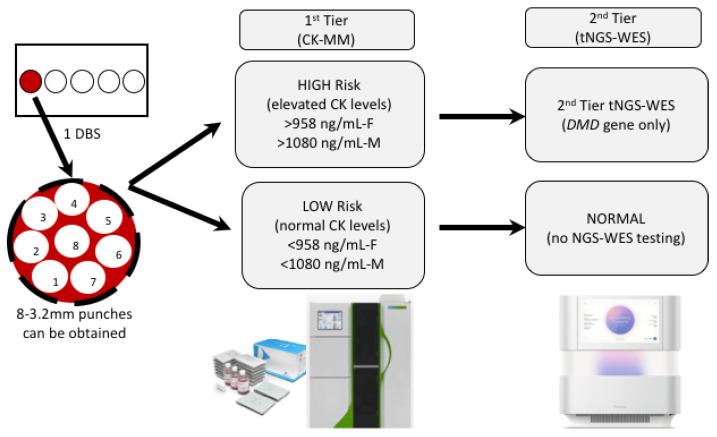
A single DBS-based two-step algorithm. The CK-MM Immunoassay is the first-tier test which uses a fixed cut-off and performed at Perkin-Elmer (Pittsburgh, PA, USA). *DMD* sequencing from NGS-WES is the second-tier test performed at NCGM, Inc. (Raleigh, NC, USA).

**Figure 3 IJNS-07-00077-f003:**
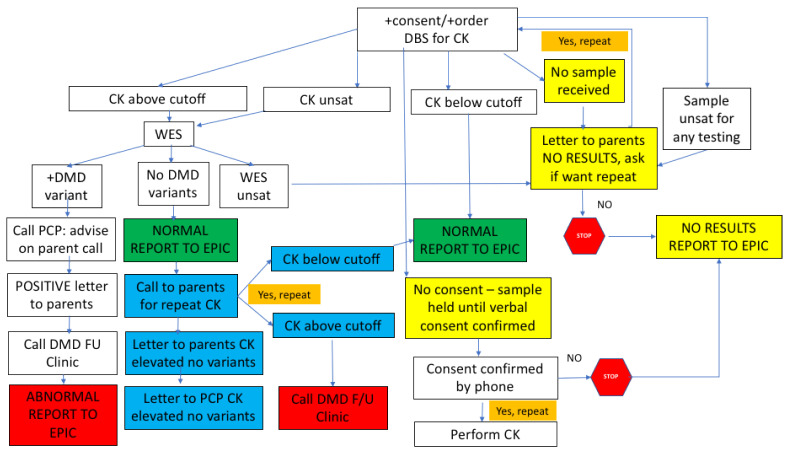
A post-analytic workflow sDMDNBS: result transmission.

**Figure 4 IJNS-07-00077-f004:**
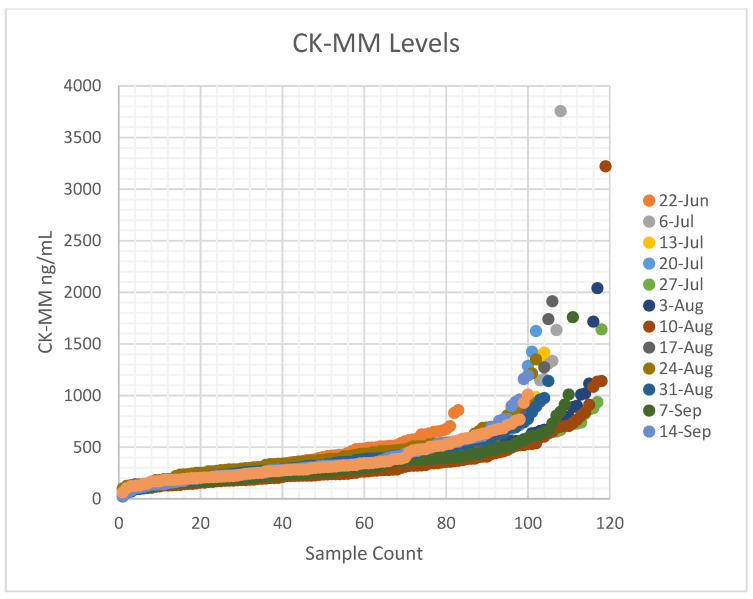
CK-MM values (Y-axis) for each sDMDNBS batch are plotted from lowest to highest levels (X-axis). The date and month refers to the week the samples were collected and processed in 2021.

**Figure 5 IJNS-07-00077-f005:**
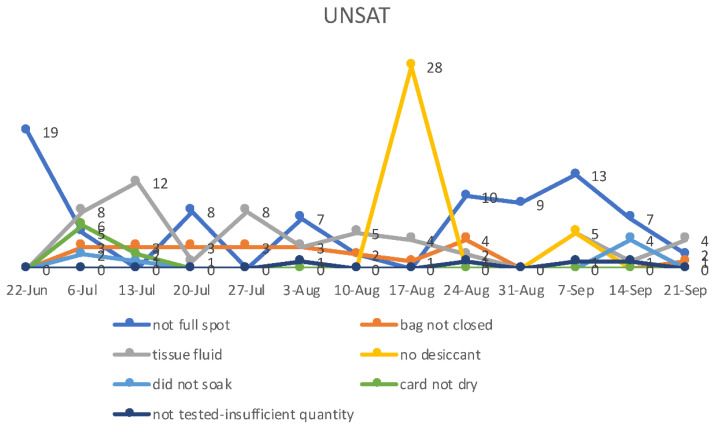
“Unsatisfactory” or UNSAT refers to a situation where screening results cannot be accurately interpreted because of a problem with insufficient quantity, blood spot collection (soaking issue or tissue fluids in spot), missing or inaccurate information on the newborn screening card, or a problem with the infant’s age of collection.

## Data Availability

The data for clinical testing are in the electronic medical records of BWH.

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
