# Peer review of "Implementation of Hospital-Based Supplemental Duchenne Muscular Dystrophy Newborn Screening (sDMDNBS): A Pathway to Broadening Adoption"

_2409-515X, 2021, doi:10.3390/ijns7040077_

Round 1
Reviewer 1 Report
In the manuscript entitled “Implementation of Hospital-based Supplemental Duchenne Muscular Dystrophy Newborn Screening (sDMDNBS): A Path To Broaden Adoption”, the authors showed a new screening strategy for DMD which combined 1st-tier CK-MM assay and 2nd-tier tNGS-WES analysis. The manuscript is well written and their protocols are clear.
However, it is difficult for me to understand the following description about the cost of testing (lines 176-180). They said, “We were aware of the importance of the cost of testing and realized that ultimately our starting costs would need to be reduced in this algorithm to become implementable for widespread public health-based DMD NBS. We initially set a target for the 1st-tier at $5/sample and the sequencing cost (at a 1% referral) to an additional $3 for a total cost of $8.” Even if labor costs are not taken into consideration, I think the total cost of $8 is incredibly inexpensive. If the authors showed the basic data (ex. number of newborn infants screened with the new system in the hospital’s area, frequency of the infants with high CK-MM, cost reduction for tNGS-WES analysis etc.), the readers could be convinced that the cost calculation was rooted in the reality.
The authors suggested the possibility that their new system can identify other diseases than DMD. They only say that "elevated CK and the WES finding" was able to benefit the newborn with a prompt diagnosis for a highly heterogeneous disease (lines 509-511).
I think, however, it is necessary to carefully discuss how to handle with such secondary findings. The diagnosis can be a pre-symptomatic diagnosis of an incurable disease with adult onset. The authors should have mentioned it in the discussion section.
Author Response
Reviewer 1 Comments:
In the manuscript entitled “Implementation of Hospital-based Supplemental Duchenne Muscular Dystrophy Newborn Screening (sDMDNBS): A Path To Broaden Adoption”, the authors showed a new screening strategy for DMD which combined 1st-tier CK-MM assay and 2nd-tier tNGS-WES analysis. The manuscript is well written and their protocols are clear.
Response: We thank the reviewer for these graceful comments on our strategy and clarity of the manuscript. We hope our work shows that this is scalable for the near term as well as longer term.
However, it is difficult for me to understand the following description about the cost of testing (lines 176-180). They said, “We were aware of the importance of the cost of testing and realized that ultimately our starting costs would need to be reduced in this algorithm to become implementable for widespread public health-based DMD NBS. We initially set a target for the 1st-tier at $5/sample and the sequencing cost (at a 1% referral) to an additional $3 for a total cost of $8.” Even if labor costs are not taken into consideration, I think the total cost of $8 is incredibly inexpensive. If the authors showed the basic data (ex. number of newborn infants screened with the new system in the hospital’s area, frequency of the infants with high CK-MM, cost reduction for tNGS-WES analysis etc.), the readers could be convinced that the cost calculation was rooted in the reality.
Response: We have made this statement more clear, that we targeted a cost of $8/baby screened, inclusive of labor. We acknowledge this point understanding that high cost could be a barrier to the initiation of such screening, but meant to imply that this $8 cost would be in-line with the costs of testing for disorders currently included in state newborn screening panels. It reads as follows:
With consideration of the importance of the cost of testing to become implementable for widespread public health-based DMD NBS, we initially set a cost target (inclusive of labor) for the first-tier CK assay at $5/sample. Second-tier sequencing costs (at a 1% referral rate and a cost of $300 per genomic test) spread over the population would add to $3 to the cost of each sample, for a total cost of $8 per infant tested.
The authors suggested the possibility that their new system can identify other diseases than DMD. They only say that "elevated CK and the WES finding" was able to benefit the newborn with a prompt diagnosis for a highly heterogeneous disease (lines 509-511).
Response: We address the reviewer comment on the use of the WES beyond DMD screening as a diagnostic test, and only with a phenotype driven request from a clinician and with consent from the parents for diagnostic analysis.
We added the following sentence:
Given that the WES sequence had already been generated on this newborn and was available for analysis, a diagnosis could be speeded and costs saved.
We also modified the following sentences in the section leading up to the above sentences:
Thus far, none of our elevated CK-MM values have resulted in the identification of a DMD variant. For neonates with an abnormal neuromuscular exam, we have made accessible to our clinicians, after obtaining expanded consent for additional WES analysis, a phenotype directed diagnostic analysis beyond DMD on an exome generated due to CK-MM elevation. One such newborn with a CK-MM value above cut-off, and reflexed for tNGS/WES testing, had a heterozygous pathogenic Chr 16 microdeletion identified. This finding would not have been detectable using pre-fixed neuromuscular tNGS panels.
I think, however, it is necessary to carefully discuss how to handle with such secondary findings. The diagnosis can be a pre-symptomatic diagnosis of an incurable disease with adult onset. The authors should have mentioned it in the discussion section.
Response: This is an important point the reviewer raises. We have added the following comment.
There is a risk that with screening for CK could lead to the identification of disorders that are not treatable. So far in NBS programs that have utilized elevated CK screening, the approach of a second CK or limiting the panel to known disorders or genotypes have reduced findings of asymptomatic or initial CK elevations due to muscle trauma during progression through the birth canal. By increasing the CK enzyme activity cut-off value to 2000 U/L, algorithms have limited the identification of diseases like LGMDs that do not have approved therapy, although typically these were identified at a much lower frequency [25]. Gene therapies in LGMDs are now entering early stages of clinical trials, and therefore both non-specific muscle preserving treatments (like steroid treatment), symptomatic treatment and disease progression halting therapies may become available. Corrective surgeries or monitoring of cardiac and pulmonary function are also important management approaches. Nevertheless, in our model, these issues are addressed by the referral to a follow-up center of any newborn whose repeat CK remains elevated in the absence of an identified DMD variant.
Reviewer 2 Report
Herein, the authors describe their exciting newborn pilot screening experience to identify Duchenne muscular dystrophy babies by means of CK measurements (1st tier) and further confirmation by tNGS of DMD locus (2nd. tier) from the original dried blood spot sample. Also the authors discuss the possibility of expand the diagnostic possibilities of NGS to WES (to identify another less common muscular dystrophies) using the same DBS sample. Very early identification of male patients with dystrophinopathies seems to be encouraged in view of expanding of new treatments. At this time, their propose seems highly justifiable and reliable. All stages (analytical, pre- an dpost-analytical phases) are sufficiently described, although I have a minor recomendations:
Woudl be interesting to include a brief description of employed NGS strategy (library construction, expected coverage, filtering, exonic deletion/duplication parameters, etc.)
Clearly statements/justification need to be included to perform Newborn screening of DMD to females newborns (even if a heterozygous female is identified: ethical concerns).
Author Response
Reviewer 2 Comments:
Herein, the authors describe their exciting newborn pilot screening experience to identify Duchenne muscular dystrophy babies by means of CK measurements (1st tier) and further confirmation by tNGS of DMD locus (2nd. tier) from the original dried blood spot sample. Also the authors discuss the possibility of expand the diagnostic possibilities of NGS to WES (to identify another less common muscular dystrophies) using the same DBS sample. Very early identification of male patients with dystrophinopathies seems to be encouraged in view of expanding of new treatments. At this time, their propose seems highly justifiable and reliable. All stages (analytical, pre- and post-analytical phases) are sufficiently described, although I have a minor recomendations:
Response: We thank the reviewer for these gracious comments.
Woudl be interesting to include a brief description of employed NGS strategy (library construction, expected coverage, filtering, exonic deletion/duplication parameters, etc.)
Response: We have added the following in the manuscript. We have also added the CNV section in this section for consistency.
For tNGS/WES, a hybrid capture is performed using a custom Twist Exome capture kit that targets coding regions, splice sites, and clinically relevant non-exonic hotspot regions with initial clinical analysis being performed on the DMD gene. Paired end sequencing is performed using the Illumina NextSeq2000 system. Reads are assembled and were aligned to reference sequences based on NCBI RefSeq transcripts and human genome build GRCh37/hg19.The average depth of coverage across the target is 100X. Approximately >99.5% of the target region is routinely covered at a minimum depth of 20X. FastQC/Fastp is used to assess the quality of the raw data. Variant calls are generated using Samtools for the Burrows-Wheeler Aligner (BWA) followed by GATK based analysis. This test detects >99% of substitution variants and >80% of small insertions and deletions up to 25 bp in length across the genes based on sequencing of NA12878 compared to reference calls from the National Institute of Standards and Technology Genome In a Bottle Consortium. Variants were interpreted in a manner consistent with the 2015 American College of Medical Genetics and Genomics and Association for Molecular Pathology recommendations and variants classified as being or uncertain significance, likely pathogenic, pathogenic and other variants as indicated are reported if related to the patient phenotype. Benign and likely benign variants are not included in this report. Given that a majority of DMD cases are due to deletion events (60-70%) we made CNV calls on the WES dataset. In addition, we do a manual check for coverage for each exon and a flag for no read or low reads. As this is a disorder associated with a gene on the X-chromosome and males have only one X-chromosome, missing CNV calls in males is unlikely. Females have a very low incidence of the disease but have a risk that a heterozygous call may be missed. We have confirmed prior multi-exonic calls in samples from affected individuals with DMD and other disorders by other molecular assays. The DNA quality is more likely to impact duplication detection, but this is a general limitation of short-read NGS. This issue may be slightly more pronounced in DBS as compared to whole blood samples. The primary reason for our confidence in our CNV calling is our dynamic baseline approach in which baselines are created from a pool of thousands of samples to match as closely as possible to the sample under analysis. This significantly improves sensitivity which is often lost in an attempt to ’normalize’ the baseline.
Clearly statements/justification need to be included to perform Newborn screening of DMD to females newborns (even if a heterozygous female is identified: ethical concerns).
Response: We have added the following in the manuscript.
- Because NBS screening programs are administered by the states and all states conduct their programs in a gender neutral or universal fashion we have defined universal screening as “screening both males and females…” to avoid confusion of the word universal in the discussion section.
- We have inserted a statement as to why we chose to offer screening for female newborns
We opted to offer sDMDNBS to the parents of all newborns regardless of sex given that DMD can, although less commonly, affect females.
- We have also inserted a statement as to why screening for female newborns is beneficial
As the LGMDs are autosomal recessive or dominant disorders, screening for females can enable identification of mutations in autosomal genes or those heterozygous for DMD mutation who are carriers and/or manifest a form of the disease, allowing this two-tier method of screening to account for subjects with elevated CK levels.
- We have also stated why screening for females is beneficial in context of how the variant arises de novo (in the Introduction section as well as in the discussion section). We hope that the value of screening females, including carriers is made in these statements.
“One third of DMD cases result from de novo mutations in maternal carriers or affected males. “
“...screening female carriers may prevent 25-30% of future cases”.
Round 2
Reviewer 1 Report
The authors have properly addressed the points raised by the reviewer.